# Oxidative Stress, Cytotoxic and Inflammatory Effects of Urban Ultrafine Road-Deposited Dust from the UK and Mexico in Human Epithelial Lung (Calu-3) Cells

**DOI:** 10.3390/antiox11091814

**Published:** 2022-09-14

**Authors:** Jessica Hammond, Barbara A. Maher, Tomasz Gonet, Francisco Bautista, David Allsop

**Affiliations:** 1Division of Biomedical and Life Sciences, Faculty of Health and Medicine, Lancaster University, Lancaster LA1 4YQ, UK; 2Centre for Environmental Magnetism and Palaeomagnetism, Lancaster Environment Centre, Lancaster University, Lancaster LA1 4YQ, UK; 3Laboratorio Universitario de Geofísica Ambiental, Centro de Investigaciones en Geografía Ambiental, Universidad Nacional Autónoma de Mexico, Morelia 58190, Michoacan, Mexico

**Keywords:** air pollution, cytotoxicity, road-deposited dust, inflammation, ultrafine particles, reactive oxygen species, transition metals

## Abstract

Road-deposited dust (RD) is a pervasive form of particulate pollution identified (typically via epidemiological or mathematical modelling) as hazardous to human health. Finer RD particle sizes, the most abundant (by number, not mass), may pose greater risk as they can access all major organs. Here, the first in vitro exposure of human lung epithelial (Calu-3) cells to 0–300 µg/mL of the ultrafine (<220 nm) fraction of road dust (UF-RDPs) from three contrasting cities (Lancaster and Birmingham, UK, and Mexico City, Mexico) resulted in differential oxidative, cytotoxic, and inflammatory responses. Except for Cd, Na, and Pb, analysed metals were most abundant in Mexico City UF-RDPs, which were most cytotoxic. Birmingham UF-RDPs provoked greatest ROS release (only at 300 µg/mL) and greatest increase in pro-inflammatory cytokine release. Lancaster UF-RDPs increased cell viability. All three UF-RDP samples stimulated ROS production and pro-inflammatory cytokine release. Mass-based PM limits seem inappropriate given the location-specific PM compositions and health impacts evidenced here. A combination of new, biologically relevant metrics and localised regulations appears critical to mitigating the global pandemic of health impacts of particulate air pollution and road-deposited dust.

## 1. Introduction

Human exposure to outdoor, fine-grained airborne particulate matter (PM_2.5_, with an aerodynamic diameter < 2.5 µm) was estimated to be responsible for an excess of 1.8 million deaths in urban areas in 2019 [1]; 99% of the world’s population is exposed to high particulate pollution levels, i.e., above the World Health Organization annual mean limits of 15 μg/m^3^ for PM_10_ (PM with an aerodynamic diameter <10 µm), and 5 μg/m^3^ for PM_2.5_ [2]. Epidemiological studies demonstrate significant associations between PM exposure and adverse health impacts, including pulmonary diseases [3], cardiovascular diseases (CVD) [4], brain tumours [5], and neurodegenerative diseases [6,7]. Road-deposited dust (RD) occurs when airborne PM, a mixture of organic and inorganic molecules from anthropogenic and natural sources, settles on/near road surfaces. RD can pose a substantial potential hazard to human health since it comprises an accumulating reservoir of deposited particulates, which can be re-suspended multiple times (e.g., through traffic-induced turbulence), providing multiple opportunities for inhalation/ingestion by all road-users and those living and/or working within close proximity to major roads. RD can further accumulate pollutants in situ, including carbonaceous compounds [8], heavy metals [9], and polyaromatic hydrocarbons (PAHs) [10].

The composition of PM, and thus of RD, is likely to vary significantly on local, national, and international scales. The cellular targets, toxic effects, and mechanisms of specific particle size fractions of PM and RD, and of PM and RD arising from different locations/sources, are currently imperfectly understood. Improved understanding of the specific, causal impacts of PM and RD, and of their differing components, would provide an evidenced rationale for legislative mitigation to reduce PM emissions; and may also be key in developing new therapeutic approaches to treat those already suffering adverse, PM- and RD-induced health outcomes.

Anthropogenic contributors to RD include not only diesel/petrol exhaust but also non-exhaust emissions (NEEs), such as brake, tyre, and road/asphalt wear, and industrial sources, e.g., combustion-derived emissions from factories, and space heating. Natural contributors include soil, endotoxins (bacteria), pollen, and aeolian dust [9,11,12,13]. As RD is often derived from diverse sources, its chemical and biological composition also varies, typically by location [13], but also with season and climate [14]. RD often contains a wide range of metal-bearing particulates, some attributable to specific sources; e.g., Ba is an additive in most brake pads [15], TiO_2_ in road paint [16], and Pt and Pd are released from catalytic converters [17]. The presence in RD of redox-active transition metals and carcinogenic compounds (e.g., PAHs) is detrimental to human health [18]; exposure to PAHs in RD was associated with an incremental lifetime cancer risk (ILCR) of 9.9 × 10^−4^ in Taiwanese adults [10] (ICLR > 10^−4^ indicates high carcinogenic risk [19]). Several studies report a significant amount of strongly magnetic, iron-rich particles, such as magnetite (Fe_3_O_4_), in RD [11,20]. Magnetite nanoparticles (MNPs) are often associated with other metal elements such as Co, Cr, Cu, Mn, Ni, Pb, Zn [12], and PAHs [21], and magnetic methods are increasingly used for monitoring of airborne PM. MNPs are potential mediators of neurodegeneration; MNPs with a striking similarity to roadside MNPs have been found in human brain tissue [22], directly associated with key pathological markers of Alzheimer’s disease (senile plaques and neurofibrillary tangles) [23,24], and may induce oxidative stress [25], leading eventually to cell death [26].

RD is estimated to comprise 25.7% of PM_10_ in Brazil [27], 55% of PM_10_ in India [28], and 24.6% of PM_2.5_ in Lanzhou, China [29]. Conventionally, air quality is monitored by measuring PM mass concentrations (typically reported as the mass (µg) of PM_10_ and/or PM_2.5_ per m^3^ air). Such mass-based metrics are usually dominated by coarser PM. Conversely, in terms of particle number concentrations, ultrafine particles (UFPs, <1 µm) are both by far the dominant fraction, and currently unaccounted for when setting PM limits/guidelines [30]. UFPs frequently represent the majority of the particles to which humans are exposed [25]. Neither RD nor NEEs are restricted currently in terms of exposure limits or emission reductions, despite their abundance (NEEs reportedly form 60% of PM_2.5_ by mass in the UK [31]), potential risk to human health, and contribution to PM when aerosolised.

Compared with larger PM size fractions, UFPs can disperse more widely in the environment [30] and their toxicity is reported to be greater [5,32], likely due to their high surface reactivity [33].

UFPs can penetrate further into the body; e.g., a multiple path particle dosimetry model suggests highest deposition of particles 10–100 nm in size in the alveolar region, regardless of their density. Alveolar deposition of larger particles (100 nm–1 µm) requires higher particle density (10 g/cm^3^). There is also relatively high deposition of nanoparticles in the tracheobronchial tract, whilst large particles (<10 µm) deposit primarily in the extra-thoracic and tracheobronchial regions [34].

UFPs may enter the body via inhalation into the lungs [35] and olfactory nerve [36], ingestion [37], and/or dermal penetration [38]. If invading microbes, and foreign bodies such as UFPs, evade the thick protective mucus layer in the lungs, epithelial cells are the first cellular line of defence (Figure 1). The epithelial cells of the human respiratory system defend against incursion of inhaled particulates, primarily via physical barriers formed by cell adhesion proteins (e.g., E-cadherin) and tight junctions (e.g., occludin). Additional defence arises through the release of chemokines, cytokines, and growth factors, and production of reactive oxygen species (ROS) and nitrogen species and antimicrobial proteins [39]. Various antioxidants (e.g., glutathione) also have a protective effect, but decrease in abundance deeper into the respiratory tract (i.e., into the regions penetrated by UFPs) [40].

Inhalation of UF-RDPs may occur when settled road-deposited dust (RD) particles become aerosolised due to wind conditions and/or traffic movement. RD contains a range of particle sizes as shown by the particle number concentration (PNC) graph (adapted from [12]). UFPs can become trapped in mucus or by cilia lining the airway epithelium but very small particles (~20–30 nm) may penetrate between and/or through cells. Such incursion is usually impeded by tight junctions between neighbouring cells, but these junctions can become damaged, and/or UFPs may increase paracellular permeability temporarily, allowing for transient opening of tight junctions and passage of particles [41]. UFPs may also pass transcellularly, travelling through epithelial cells lining the lungs, and can interact/damage cellular components during this passage [42]. The impact of UFP inhalation is not limited to the lungs but extends systemically via three possible mechanisms [43]: 1. incursion into the circulatory system, with subsequent travel to and deposition in extra-pulmonary organs, e.g., [36,44,45]; 2. stimulation of the release of pro-inflammatory mediators from the lungs, which enter the circulation and affect tissues beyond the lungs, e.g., [46]; and/or 3. interaction of pollutant UFPs with the nerves/receptors in the lung, which activates the autonomic nervous system (ANS) to affect a systemic change or response in the body [47]. Particles <200 nm can be transported to the brain directly [22,37,48] via the olfactory bulb [36], and may also be transported to the central nervous system (CNS) via other nerve pathways (trigeminal, vagus, neuroenteric).

UFP cytotoxicity is likely to arise via different pathways, including oxidative stress, damage to the cell membrane, altered gene expression, mitochondrial dysfunction, and/or DNA damage, including the inability to repair this damage [34]. Excess oxidative stress can cause a hierarchical oxidative response in cells: exposure to PM/RD generates the production of free radicals and/or ROS (e.g., via the Fenton reaction); altering the oxidant/antioxidant balance within the cell and stimulating expression of cytoprotective proteins. If the stress is prolonged and/or chronic, secretion of cytokines (e.g., IL-6, IL-8) is triggered, inducing an inflammatory response [49]. If the oxidative stress does not subside, the cell can undergo death via apoptosis or necrosis [50].

Since they are the first cells to encounter inhaled particulates, human lung epithelial cells have been used to model in vitro toxicity of air pollution, including RD particles, brake wear particles, tyre wear debris, and exhaust emissions, e.g., [14,51,52,53,54,55]. Cytotoxicity studies investigating RD using human cell lines are sparse [18,54,55,56,57,58,59] (Appendix A); with the majority of studies examining the effects of particles ≤2.5 µm. Exposing liver (HepG2) and skin (KERTr) cell lines to 66.7µg/100µL RD from Guangzhou, China decreased cell viability after 72 h; by 53.9% and 71.4%, respectively [54]. The decrease in viability was correlated to the sum of total metal (loids) present in the sample, with Zn, Mn, Cu, and Ni identified as major components [54]. The cytotoxicity of UF, traffic-related pollution particles has also been investigated in rat cell models [60,61]. Little is known about the biological effects in human cells of the UF fractions of RD, which are potentially hazardous owing to their small size, varied, often metal-rich, composition, and abundance in the environment.

To our knowledge, we report here the first investigations of the effects of the ultrafine fraction of RDPs in vitro. The aims of this study were to: (1) extract and characterise the ultrafine fraction (<220 nm) of road-deposited dust particles (RDPs) collected at heavily trafficked sites [62,63,64] in three contrasting cities; (2) examine the oxidative, cytotoxic, and pro-inflammatory responses of human bronchial epithelial (Calu-3) cells treated with UF-RDPs from these different locations; and (3) compare the cellular effects induced by UF-RDPs from these three cities; namely, a small UK city (Lancaster) and larger UK city (Birmingham), compared with the more highly polluted Mexico City.

## 2. Materials and Methods

### 2.1. Sampling Sites

The UK RD samples were collected within 0.5 m of heavily trafficked roads: the A6 at Cable Street (Lancaster, UK) [11] and A38, close to the Bristol Road Observation Site (Birmingham, UK). The Lancaster site is located near a taxi rank and opposite a bus station, where traffic queues are frequent. The Birmingham site is located close to two busy, traffic-lighted junctions, near the University of Birmingham entrance. Mexico City dust was collected from an area of 1 m^2^ of the Constitución de la República Avenue. Sample site information is summarised in Table 1. The sampling sites were selected in order to compare sites with a range of PM_2.5_ concentrations, within the same country (UK), and to compare to a densely populated city with greater air pollution levels (Mexico City, Table 1). We have previously reported the presence of UF magnetic nanoparticles (likely originating from PM, including RD) in human brains from Mexico City, and northern England [22,37,48], and magnetically characterised RD from Lancaster and Birmingham [11]. The present work builds on this prior RD characterisation and explores the effects of UF-RDPs (which include UF magnetic particles) at the cellular level in order to investigate the possible consequences of the presence of such particles in human organs including the brain and heart [22,44].

### 2.2. Ultrafine Particle Extraction from Road-Deposited Dust

The bulk dust samples were dispersed (via sonication) in triple-filtered 100% ethanol and filtered through multiple (at least 3) 0.22 µm polyether sulphone (PES) filters based on a protocol as per [51]. A filter pore size of 0.22 µm was used in order to provide sufficient material for repeat cellular analyses. After ethanol evaporation, the concentrated particles (<220 nm, hereafter ‘UF-RDPs’) were weighed in a room with controlled temperature (20 °C) and humidity (50%), with a Mettler AT250 balance (accuracy 0.00001 g). To form stock solutions, the UF-RDPs were sonicated and re-suspended in 0.5% triple filtered bovine serum albumin (BSA) in dH_2_O [68]. Each filtered sample originated from a single original bulk RD sample.

### 2.3. Inductively Coupled Plasma (ICP) Mass Spectrometry (MS) and Optical Emission Spectroscopy (OES)

The metal content of the UF-RDP samples was quantified by ICP analyses, after acid digestion, at the University of Edinburgh. A subsample of filtered UF-RDPs was taken from each stock and the ethanol fully evaporated. Dried samples were weighed in savillex Teflon vessels and digested overnight (100 °C) in digestion mixture (3 mL HNO_3_, 2 mL HCL, 0.5 mL HF, all double distilled). After complete digestion and evaporation, samples were acidified in 2–5% ultrapure HNO_3_ then analysed for metals and elements of interest using an Agilent Varian Vista Pro (ICP-OES) or Attom Nu (ICP-MS) with the following settings: analysis mode = deflector jump, dwell time/peak = 1 millisecond, number of sweeps = 500, number of cycles = 3, resolution = 300.

### 2.4. Superconducting Quantum Interference Device (SQUID) Magnetometry

Magnetic methods are non-destructive analyses (see SI for more detailed explanation) that have been used to identify combustion- and friction-derived magnetic nanoparticles in human brain [22,37] and heart tissue [44], as well as to characterise RD and brake wear particles [11]. The magnetic content of the bulk (unfiltered) RDPs was measured with a 2G RAPID cryogenic magnetometer (at the Centre for Environmental Magnetism & Palaeomagnetism, Lancaster University, Lancaster, UK) by imparting an isothermal remanent magnetisation (IRM) at 1 Tesla (T) (using a Newport Instruments electromagnet) at room temperature (Appendix A). To identify the presence of ultrafine (~20–30 nm) magnetic particles, low-temperature magnetic measurements were made on the extracted UF-RDPs, using an MPMS3 SQUID magnetometer (Quantum Design, San Diego, CA, USA) at the Department of Physics, University of Cambridge. For UF-RDPs, IRM was imparted at 1T and 300 K, and was measured upon cooling to 10 K (at average rate 5 K/min) at the University of Cambridge. Then, IRM was imparted again (at 1 T and 10 K) and measured upon heating to 300 K. To increase signal-to-noise ratio, 10 DC scans were used for IRM at 300 K and 10 K (Appendix A).

### 2.5. Endotoxin Quantification

Endotoxin concentrations of stock solutions were determined via a quantitative kinetic limulus amoebocyte lysate (LAL) assay kit using the manufacturer’s protocol (Thermo Scientific™, Loughborough, UK).

### 2.6. Cell Culture

All cell culture reagents were purchased from Lonza Ltd. (Basel, Switzerland) unless stated otherwise. Human lung epithelial cells (Calu-3, ATCC HTB-55™) were selected due to their previous characterisation and study as targets of airborne particulate matter [51,53,69,70,71]. The immortalised cell line originated from a 25-year-old white male with lung adenocarcinoma. Calu-3 cells were maintained (until passage 20) in Eagle’s minimum essential medium (EMEM) supplemented with 10% (*v/v*) filter sterilised foetal bovine serum (FBS) (Gibco™, Thermofisher), 1% (*v/v*) non-essential amino acids, penicillin (50 units/mL), and streptomycin (50 µg/mL) at 37 °C, 5% CO_2_. Fluorescent and absorbance readings were conducted using a Tecan Infinite M200 pro spectrophotometer.

### 2.7. MTS Assay

Calu-3 cells were seeded at 40,000 cells/well in a 96-well plate and left overnight to adhere. UF-RDPs at concentrations of 0–300 µg/mL (equivalent to 0.94–94 µg/cm^2^) were prepared via sonication in UltraMem supplemented with penicillin (50 units/mL) and streptomycin (50 µg/mL). Dose and exposure times were based on a previous study with Calu-3 cells and brake wear particles [51]. BSA, used here as a stabilising agent [68], was present at equal concentrations across each set of test conditions; the observed biological responses are values normalised to BSA-exposed controls. Following a 24 h exposure, cells were rinsed with Dulbecco’s phosphate buffered saline (DPBS) and subjected to a 3-(4,5-dimethylthiazol-2-yl)-5-(3-carboxymethoxyphenyl)-2-(4-sulfophenyl)-2H-tetrazolium (MTS) assay, with the protocol based on a WST-1 assay as per [72].

### 2.8. Reactive Oxygen Species (ROS) Assay

Calu-3 cells, seeded as above, were incubated with 25 µM of the fluorescent cellular probe chloromethyl derivative of 2′,7′-dichlorodihydrofluorescein diacetate (CMDCFH2DA) (Invitrogen™, Waltham, MA, USA) in DPBS for 45 min, prior to particle exposure, at 37 °C, 5% CO_2_ [51]. Then, 100 µM tert-butyl hydroperoxide (TBHP) was used as a positive control. Fluorescence (excitation 495 nm, emission 529 nm) was measured at 0.5, 1, 2, 3, and 4 h. The 4 h timepoint was chosen to reflect the rapid clearance of UFPs from the lungs, from as little as 4 h [36]. Background controls (*n* = 3) consisting of UF-RDPs or TBHP in UltraMem were measured alongside the cell treatments for intrinsic fluorescence and subtracted from the experimental cell fluorescence readings.

### 2.9. Cytokine ELISAs

Following 24 h exposure to UF-RDPs (as above), the media samples were collected, centrifuged (15000 RCF, 15 min, 4 °C), and analysed for IL-6 and IL-8 concentrations via ELISAs conducted according to the manufacturer’s protocol (IL-6 BioLegend, London, UK, IL-8 Invitrogen™, Waltham, MA, USA). Cytokine concentrations were calculated from standards fitted using a four-parameter logistic curve-fit with program MyAssays (http://www.myassays.com/, accessed on 13 August 2021) (see Appendix A). UFPs have a large surface area for cytokine adsorption [73] which can cause interference with ELISA results. Here, known cytokine standards were spiked with UF-RDPs and measured via ELISA. Values were within 5–10% of unspiked samples measured in parallel (data not shown), suggesting any adsorption has limited impact on the assay. Data were normalised to the control, however, the raw values (ng/mL) can be seen in Appendix A.

To account for potential changes in cell number, the ELISA data were adjusted using Equation (1):(1)$$\left(\frac{IL\text{-}6\;fold\text{-}change\;relative\;to\;control}{MTS\;fold\text{-}change\;relative\;to\;control}\right)\times 100.$$

### 2.10. Statistical Analysis

All experimental results represent 3–4 individual experiments. Each set of UF-RDPs was tested independently under the same experimental conditions using identical control conditions in each case. Data were normalised to the control from the independent experiment to allow for comparison across experiments. Data are presented as mean ± standard error of the mean (SEM). Statistical analysis was conducted using SPSS 24 (IBM). Normality tests were performed using the Shapiro–Wilk test. A one-way analysis of variance (ANOVA) (with Dunnett’s post hoc) was performed to compare particle treatments with unexposed control. Comparison of location and concentration was assessed by a two-way ANOVA (with Bonferroni correction post hoc). Statistical significance levels used are: *, *p* ≤ 0.05; **, *p* ≤ 0.01; ***, *p* ≤ 0.001; ****, *p* ≤ 0.0001 where * may be substituted for ^, # or ₀ depending on the city or comparison being made.

## 3. Results

In terms of metal compositions of the three sets of UF-RDPs, it is notable that nearly all metals analysed are most abundant in the Mexico City UF-RDPs (18 out of 24), compared to the UK samples (Figure 2; Appendix A). For example, mass concentrations of Cu and Fe are 67.2 ppm and 77.1 ppm for Mexico City, followed by 29.1 ppm and 55.9 ppm for Lancaster and 19.7 ppm and 14.7 ppm for Birmingham. Conversely, Na is most abundant in the Lancaster UF-RDPs (105,489 ppm), followed by Mexico City (42,163 ppm) and Birmingham (1003 ppm). Pb concentrations are 8.0 ppm for Lancaster, 1.2 ppm for Mexico City, and 0.7 ppm for Birmingham. Cd was the only analysed element occurring in the greatest concentrations in the Birmingham UF-RDPs (6.1 ppm), followed by Lancaster (1.1 ppm) and Mexico City (below detection limit).

In terms of their magnetic content, the measured mass concentration of magnetite in the Mexico City UF-RDPs was ~0.24–0.79 wt.%. The presence of a broad Verwey transition identifies the presence specifically of magnetite (a mixed Fe^2+^/Fe^3+^ iron oxide) with ultrafine (~20–30 nm) magnetite particles evident from the large (~43%) increase in remanence at low temperature (10 K) compared to that at 300 K (Appendix A) [74,75]. Due to the low sample mass extracted, the magnetic content of the Lancaster and Birmingham UF-RDPs was unmeasurable; however, IRM data for the bulk samples can be seen in Appendix A.

Detectable but minor levels of endotoxin were present in all three UF-RDP samples (8.75–9.25 EU/mg), as assessed by LAL assay (Appendix A).

Calu-3 cells were exposed to UF-RDP doses of between 0 and 300 µg/mL (0.94–94 µg/cm^2^). It is noteworthy that significant cellular responses were elicited even at low and intermediate UF-RDP doses, especially for the Mexico City UF-RDPs. Although the maximum dose is larger than any single typical environmental exposure, such high doses may be indicative of responses elicited due to the chronic, repeated, and accumulated exposures to which urban dwellers are subjected in their life-course.

Following 24 h exposure to UF-RDPs (0–300 µg/mL), Calu-3 cell viability varied significantly by sample location. For the Lancaster UF-RDPs, cell viability increased, by 25–35% (Figure 3), in a dose-dependent manner (15 µg/mL upwards), similar to our previous studies with synthetic magnetite nanoparticles (unpublished data). Treatment with the Birmingham UF-RDPs caused a significant decrease in cell viability but only at the highest exposure dose (300 µg/mL, 65% decrease). In contrast, Calu-3 cells were most sensitive to the Mexico City UF-RDPs where a dose-dependent decrease in viability was seen, even at the lowest dose (3 µg/mL, 12% decline), up to a 30% decline at 300 µg/mL.

Elevated ROS was observed from the 30 min time point for all three cities (Appendix A). At the 4 h timepoint, Mexico City UF-RDPs were the most potent, stimulating increased ROS from 75 µg/mL. Birmingham UF-RDPs induced little increase in ROS generation except at the maximum dose (300 µg/mL, 120%), when cell viability also showed maximum decline (Figure 4). Lancaster UF-RDPs were least potent in terms of ROS generation.

The release of IL-6 and IL-8 cytokines following 24 h exposure to UF-RDPs was quantified by ELISA (Figure 5). To account for potential changes in cell number, the ELISA results (Figure 5) were adjusted using the MTS data (Figure 3). Unadjusted data and absolute values in ng/mL are given in the Appendix A and generally show similar trends to unadjusted data at non-lethal doses. Following exposure to the extracted UF-RDPs, a dose-dependent increase in IL-6 release was observed in response to doses of 75 µg/mL or above from all three cities, with the greatest increase in release (1648%, corresponding to an increase of 8.3 ng/mL) in response to 300 µg/mL Birmingham UF-RDPs (Figure 5A). An overall dose-dependent increase in IL-8 was observed for Mexico City and Birmingham UF-RDPs, and to a lesser extent for Lancaster UF-RDPs (Figure 5B). For Mexico City UF-RDPs, IL-8 release peaked at 75 µg/mL and then declined at the higher doses, possibly due to high cytotoxicity at these doses. The largest increase in both IL-6 and IL-8 was stimulated by the Birmingham UF-RDPs (adjusted data, 1648%/increase of 8.3 ng/mL and 408%/increase of 12.3 ng/mL, respectively). Compared with IL-6, IL-8 release displayed significantly greater variation with city source.

## 4. Discussion

The increased viability resulting from the Lancaster UF-RDPs may represent an increase in cell number (proliferation) or metabolic activity (Figure 3). The Pb content of the Lancaster UF-RDPs—sampled close to the city bus station—is notably high, ~8 x higher compared with those from Birmingham and Mexico City. Cell proliferation might thus reflect the tumorigenic effect of Pb [76]. Pb was reportedly 4 x higher in settled bus dust relative to background soil [77]. Proliferation might also reflect replenishment of damaged epithelial cells, and/or airway remodelling [52]; i.e., changes in the composition, structure, or thickness of (structural) elements of the airway. In response to high PM exposure, lungs from female life-long residents of Mexico City displayed extensive airway remodelling including formation of fibrous tissue (pulmonary fibrosis) [78]. Conversely, lead oxide nanoparticles have been implicated in the induction of apoptosis following mitochondrial damage [79]. Lead nanoparticles, however, did not induce apoptosis in A549 lung cells [80], so the apoptotic effect may be specific to the type of lead nanoparticle and/or cell type, and in the case of PM is also likely influenced by other compounds and elements present in the heterogenous mixture. It was not possible to assess cell proliferation due to the limited sample material available, but future work would usefully include the assessment of cell proliferation, for example, via trypan blue.

In contrast, the Mexico City UF-RDPs induced a dose-dependent decrease in cell viability, reflecting reduced metabolic activity or reduced proliferation, and/or cell death. This decreased viability likely reflects the abundance of metals in these UFPs, including Fe, Zn, Mn, Pb, Cu, Cr, and Ni [12,13]. Transition metals can catalyse ROS production in situ via the Fenton reaction, leading to oxidative damage to lipids, DNA, and proteins, and eventually cell death [81]. The majority of the analysed metals occur in greatest concentrations in the Mexico City UF-RDPs (Figure 2; Appendix A). Of these metals, Ba, Co, and Ni may have the strongest influence on the observed cytotoxic response. This is because the other metals (Cr, Cu, V, and Zn) which are most abundant in the Mexico City UF-RDPs are present at higher concentrations in Lancaster UF-RDPs than Birmingham UF-RDPs, yet there was no decline in cell viability in response to Lancaster UF-RDPs compared to Birmingham (Figure 2 and Figure 3; Appendix A). Ni may be of particular importance, as it is present in the Birmingham and Mexico City UF-RDPs but not in the Lancaster particles (Figure 2; Appendix A); Ni in RD from South Korea was correlated with cytotoxicity [56]. Birmingham UF-RDPs only decreased Calu-3 viability at the highest dose (300 µg/mL), when antioxidant defences likely were overwhelmed.

At the maximum dose (300 µg/mL), intracellular ROS levels were elevated within 30 min of exposure to the UF-RDPs from all three cities. Lancaster UF-RDPs were least potent in terms of ROS generation; in contrast, higher ROS concentrations were observed following Birmingham and Mexico City UF-RDP exposures (Figure 4). Together, these data suggest that the lower ROS levels induced by Lancaster UF-RDPs could stimulate proliferation (increased cell viability), whereas the higher ROS levels induced by Mexico City (and Birmingham) UF-RDPs result in cell death via oxidative damage. Some similar responses have been reported for RD samples ≤2.5 µm and ≤10 µm. Up to 180% increased ROS production was observed in human corneal epithelial cells after 24 h exposure to RD from residential areas of the city of Gangdong-Gu, Korea [59]. Re-aerosolised RD_2.5_ from 10 Chinese cities displayed correlation between cellular ROS production and heavy metal concentrations (Cr, Mn, Zn, Ni, Pb) [55].

In our UF-RDPs, ultrafine magnetite particles (~20–30 nm) were abundant in the Mexican sample, and we have previously detected magnetite/maghemite in bulk RD at the same sampling sites in Lancaster and Birmingham [11]. Given the catalytic role of Fe (especially Fe^2+^) in the Fenton reaction, ultrafine magnetite may play a particular role in the dose-dependent increases in ROS generation seen here [25].

Dose-dependent increases in IL-6 and IL-8 were observed in response to UF-RDPs from all three cities (Figure 5). Toxic concentrations of UF-RDPs may result in cell death, which has been associated with an increase in IL-6 secretion [82], so examination of sublethal concentrations (i.e., viability <80% [58]) is important. Using this criterion, none of the Lancaster UF-RDP doses were lethal, whereas for Mexico City UF-RDPs, 15, 30, 75, and 300 µg/mL doses (corresponding to viability of 78, 75, 75, and 70%, respectively) were lethal. For the Birmingham UF-RDPs, only the maximum dose (300 µg/mL) was lethal. Excluding these data and focusing on sublethal doses, elevated IL-6 and IL-8 release was observed in response to UF-RDPs from all three cities from a relatively low dose (e.g., 15 µg/mL for IL-8). The largest increases in cytokine release were in response to the Birmingham UF-RDPs (Figure 5). Interestingly, Puisney et al. (using the same cell line and methods) report no change in IL-8 secretion in response to 0–300 µg/mL doses of UF brake wear particles [51] but an increase (up to ~350%) in IL-6 secretion. These differences in observed biological response may thus reflect differences in the PM samples tested—UF, dynamometer-derived brake wear [51]—compared to our UF-RDPs. The elements most abundant in the dynamometer-derived brake wear particles were Fe and Cu, followed by Si, Al, and Zn, whereas our UF-RDPs are dominated by Na and Ca, followed by Zn, K, and Mg. Roadside PM contains not only traffic- and industry-derived compounds, but also naturally derived elements/metals, including Al, Ca, Fe, K, Mg, or Na [83]. Some of the naturally (soil-)derived compounds might also be involved in the Calu-3 biological responses. Alternatively, it is possible that transition metals increase ROS/oxidative damage, decreasing cell viability, and also trigger an inflammatory response in the form of increased IL-6 and IL-8 secretion (Figure 6) [84,85].

RD is, of course, a heterogenous mixture, and other components (not measured here) such as the organic fraction (e.g., carbonaceous compounds, PAHs) are also known to be cytotoxic [19], induce ROS generation [55], and alter inflammatory responses [86]. It is improbable, however, that (semi-)volatile compounds are responsible for the cellular responses observed here since most of these would have been lost in the particle extraction procedure (filtration in ethanol and evaporation). Endotoxins are also present in RD. Endotoxins—pyrogenic molecules shed from Gram-negative bacteria which are ubiquitous in the environment—are known to induce ROS production, stimulate inflammatory responses in humans, and are toxic at high levels [87]. Endotoxins were present albeit at low levels (8.75–9.80 EU/mg) in all three UF-RDP samples (Appendix A). Environmental endotoxins are primarily associated with the coarse (2.5–10 µm) fraction of PM, and have been implicated in ROS generation [88]. It is possible the endotoxins present in our UF-RDPs contribute to ROS generation and pro-inflammatory responses; nanoscale endotoxin can penetrate to the alveoli and poses a higher risk to human health than its larger counterparts [89]. However, exposing Calu-3 cells to 10 µg/mL LPS (5000 EU/mL) for 4 h showed no change in ROS generation (data not shown), suggesting that the presence of environmental endotoxins at the concentrations measured did not contribute to the ROS generation observed at this timepoint in these cells.

The different biological responses seen here in response to UF-RDPs from the three cities likely derive from differences in UF-RDP physicochemical compositions, and which reflect the biogenic and anthropogenic activities in the surrounding area [13], as well as current air pollution regulations. ROS generation is generally higher in response to PM in the developing world; for example, PM from Beirut (more permissive regulations) was more redox-active than PM from Los Angeles, and Ni and V (tracers of fuel oil combustion) were enriched 5–6 × in PM_0.25_ from Beirut compared to Los Angeles [40]. Here, we see ~8 × higher Ni and 2–5 × higher V in Mexico City compared to the two UK cities. All three samples were collected from heavily trafficked road sites (Table 1), where many metals are emitted from traffic-related sources. Ba, Cu, and Fe (most abundant in Mexico City UF-RDPs) appear co-associated, and likely originate from the same source, i.e., from brake wear (e.g., [90]). Zn, also most abundant in Mexico City, has often been reported in tyre wear PM emissions [9,91]. A recent study in Toronto, Canada, observed similar co-associations between traffic-derived metals in RD, including Zn, Mo, Cr, Sn, Pb, and Ba [20]. All these metals (along with the remaining transition metals) likely contributed to the strong oxidative response observed in the lung cells exposed to UF-RDPs from the three cities studied here (Figure 6).

One limitation of this study is the difference in sampling collection time (year and season) (Table 1). Variable biological responses may reflect the season of sampling, associated with varying ambient PM sources [14]. Finally, because of the low mass of extractable UFPs, particle number concentrations were not analysed for the three samples; nor was the magnetic content of the UFPs measurable for the UK samples. To enhance further our understanding of the impacts of RD from different locations, with different physicochemical properties, future work would usefully identify the particle size distribution, morphology, and any possible particle changes within the biological media used.

Due to its abundance, reactivity, and pervasiveness at/near roadsides, growing attention is focused on the health impacts of ultrafine PM and RD. The differential oxidative stress, inflammatory, and cytotoxic responses to UF-RDPs we have observed in human lung epithelial cells represent processes which may be pathogenic if exposures to UF-RDPs are prolonged and/or chronic. We have examined lung cells but the effects of exposure to UF-RDPs extend well beyond the lung (Figure 1). For example, ultrafine magnetite nanoparticles, like those present in our Mexico City UF-RDPs, have been found in the human brain [22]. Although a direct inhalation route via the olfactory bulb is possible and likely, our measured brain and road-deposited magnetite concentrations [11,48], coupled with modelled olfactory deposition rates [92], indicate that inhalation and circulatory transport must dominate CNS translocation of such particles. Similarly, exogenous metal-rich nanoparticles have been found in the heart [25,44] and the placenta [93]. Thus, any consequences of exposure to UF-RDPs (including iron-rich, magnetic nanoparticles) are most likely to be systemic.

## 5. Conclusions

The first in vitro exposure of human lung epithelial (Calu-3) cells to the ultrafine fraction of road dust from three contrasting cities (Lancaster and Birmingham, UK, and Mexico City, Mexico) resulted in differential oxidative, cytotoxic, and inflammatory responses. Even at low and intermediate UF-RDP doses, significant cellular responses were elicited, especially by the Mexico City UF-RDPs. Given that any (semi-)volatile components are likely to have been lost during the particle extraction process, variations in the solid, metal content of the three sets of UF-RDPs are likely to be linked with the different observed cell responses.

Airborne PM is subject to regulations which rely currently on mass-based metrics such as PM_10_ and PM_2.5_, rather than particle number/composition, while RD is not currently limited/regulated by any legislation. Such mass-based metrics cannot capture the differential biological impacts induced by roadside particles as observed here. We observed a stronger cytotoxic response from Mexico City UF-RDPs compared to the UK cities, and a stronger inflammatory response from Birmingham UF-RDPs. It is therefore illogical to apply the same PM limits across different cities, where exposure to a mass- or even number-based PM limit in one city may have far worse consequences for health than exposure to the same limit in a different city. A combination of new, biologically relevant metrics, identification of specific components (e.g., metals) of RD that cause toxic effects, and localised regulations each appear critical to mitigating the global pandemic of the health impacts of particulate air pollution and road-deposited dust. A metric that provides an intermediate measurement, such as the lung-deposited surface area (LDSA) which utilises particle size distribution to estimate the surface area concentration of particles that deposit in the lung alveolar region, might be more appropriate [94]. Not only particle number but also physicochemical composition, and/or specific elemental components (e.g., transition metals), likely require regulation in order to achieve substantive mitigation of the human health impacts of exposure to RD and airborne PM.

## Figures and Tables

**Figure 1 antioxidants-11-01814-f001:**
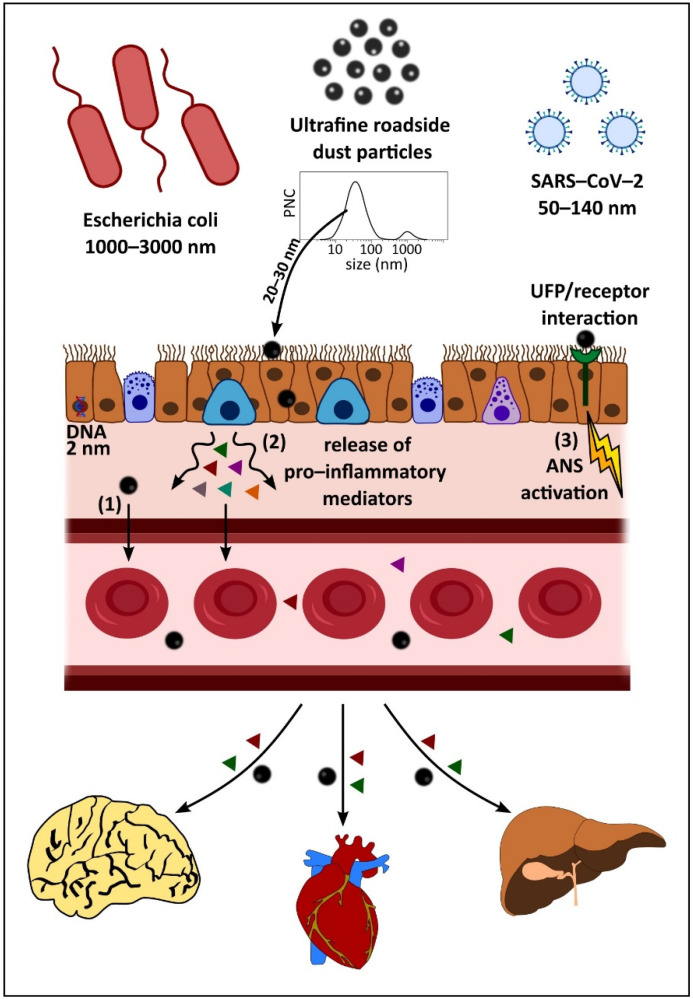
Fate of inhaled ultrafine road-deposited dust particles (UF-RDPs) in the human body.

**Figure 2 antioxidants-11-01814-f002:**
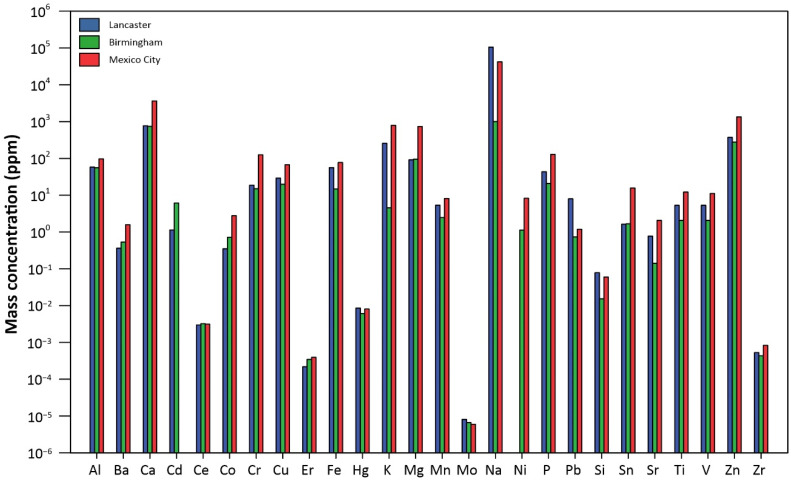
Elemental composition of the UF-RDPs from Lancaster, Birmingham, and Mexico City.

**Figure 3 antioxidants-11-01814-f003:**
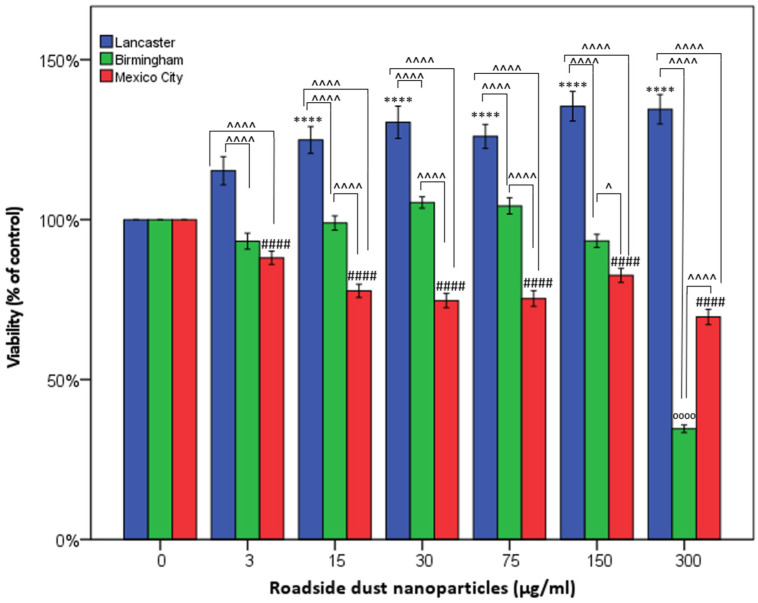
Cytotoxicity of <220 nm-sized road-deposited dust particles (UF-RDPs) on Calu-3 cells. Calu-3 cells were exposed to UF-RDPs (0–300 µg/mL) from Lancaster and Birmingham (UK) and Mexico City (Mexico) for 24 h and subjected to an MTS assay, which reflects cell viability. A one-way ANOVA with Dunnett’s post hoc was conducted, comparing treated cells to the untreated control (* Lancaster, ₀ Birmingham, # Mexico City) and two-way ANOVA with Bonferroni correction to compare the impacts of the UF-RDPs from the three different locations (^). Statistical significance levels used are: *, *p* ≤ 0.05; ****, *p* ≤ 0.0001 where * may be substituted for ^, # or ₀ depending on the city or comparison being made.

**Figure 4 antioxidants-11-01814-f004:**
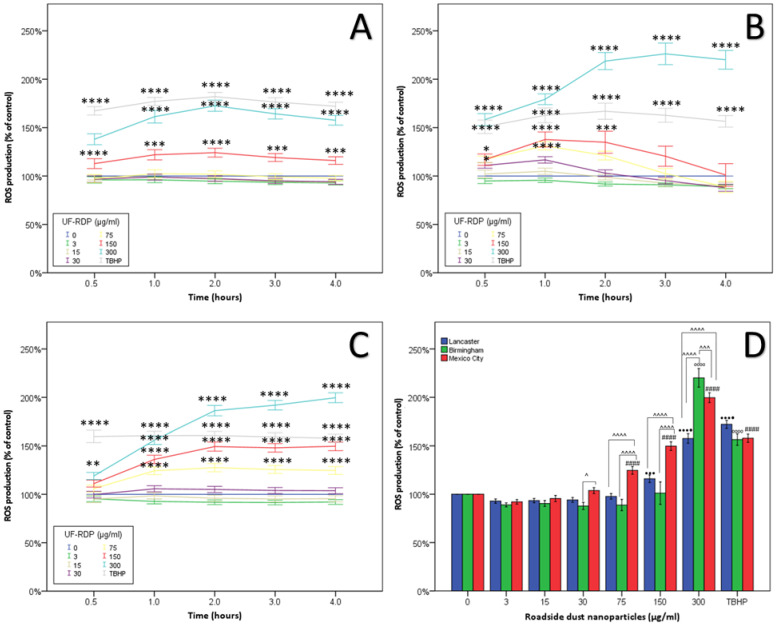
Oxidative stress in Calu-3 cells induced by <220 nm-sized road-deposited dust particles (UF-RDPs). Calu-3 cells were loaded with the ROS probe CM-DCFH-DA and exposed to UF-RDPs (0–300 µg/mL) from Lancaster (**A**) and Birmingham (**B**) (UK) and Mexico City (Mexico) (**C**). Generation of ROS was measured after 0.5 h, 1 h, 2 h, 3 h, and 4 h (**D**) exposure. Tert-butyl hydroperoxide (TBHP) was used at 100 µM as a positive control. A one-way ANOVA with Dunnett’s post hoc was conducted, comparing treated cells to the untreated control (* for A–C and for D: • Lancaster, ₀ Birmingham, # Mexico City) and two-way ANOVA with Bonferroni correction to compare the impacts of the UF-RDPs from the three different locations (^). Statistical significance levels used are: *, *p* ≤ 0.05; **, *p* ≤ 0.01; ***, *p* ≤ 0.001; ****, *p* ≤ 0.0001 where * may be substituted for ^, #, • or ₀ depending on the city or comparison being made.

**Figure 5 antioxidants-11-01814-f005:**
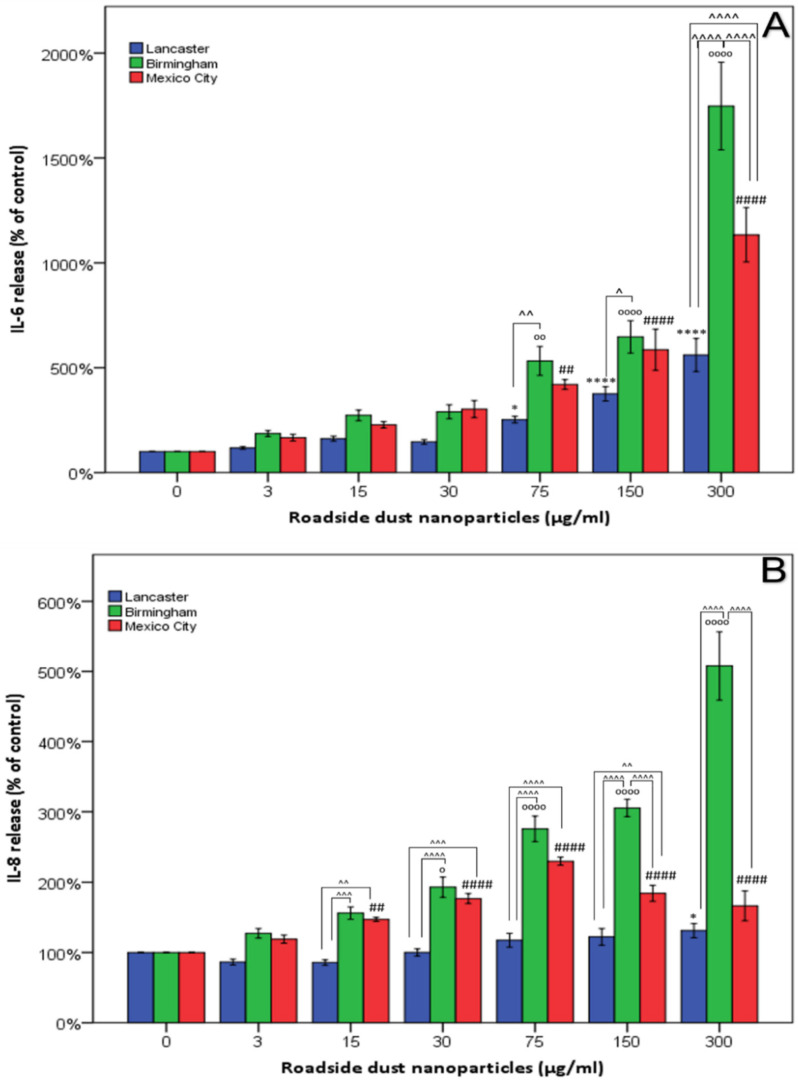
Release of pro-inflammatory cytokines in Calu-3 cells treated with <220 nm-sized road-deposited dust particles (UF-RDPs). Calu-3 cells were exposed (0–300 µg/mL) to UF-RDPs from Lancaster and Birmingham (UK) and Mexico City (Mexico) for 24 h. IL-6 (**A**) and IL-8 (**B**) concentrations in the media were quantified via ELISA. Data were adjusted using MTS (cell viability) data from Calu-3 cells treated in the same manner. A one-way ANOVA with Dunnett’s post hoc was conducted, comparing treated cells to the untreated control. (* Lancaster, ₀ Birmingham, # Mexico City) and two-way ANOVA with Bonferroni correction to compare the impacts of the UF-RDPs from the three different locations (^). Statistical significance levels used are: *, *p* ≤ 0.05; **, *p* ≤ 0.01; ***, *p* ≤ 0.001; ****, *p* ≤ 0.0001 where * may be substituted for ^, # or ₀ depending on the city or comparison being made.

**Figure 6 antioxidants-11-01814-f006:**
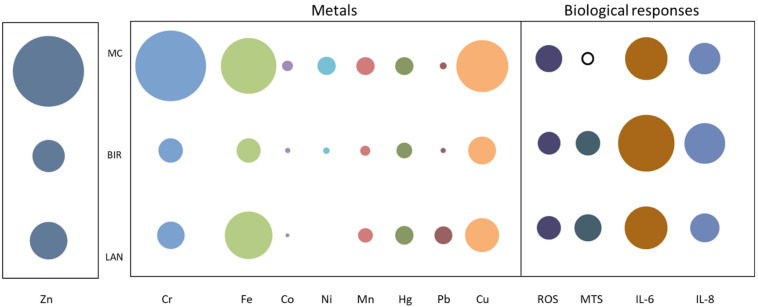
Summary of biological responses and heavy metal concentrations of UF-RDPs from Lancaster, Birmingham, and Mexico City in Calu-3 cells. The bubble plot depicts the relative abundance of selected metals (ppm) in UF-RDPs from Mexico City (MC), Birmingham (BIR), and Lancaster (LAN). The concentration of Zn is ~10-fold greater than the other metals so has been scaled separately. The biological responses to 75 µg/mL UF-RDPs are represented as percentages of the untreated control, measured at 24 h (MTS, IL-6, and IL-8) or 4 h (ROS). Solid black outline indicates a decrease, whilst filled circles indicate increases.

**Table 1 antioxidants-11-01814-t001:** Road-deposited dust sample collection sites. * The Cable Street monitoring station in Lancaster started measuring PM_2.5_ from October 2020. An approximate average for Lancaster PM_2.5_ is 8 µg/m^3^ based on data available at http://www.ukairquality.net/ (accessed on 5 November 2021). PM data obtained from [62,63,64], population data from [65,66,67].

City	Population Size (2018)	Site	Traffic (Vehicles/Day)	Date Collected	Avg. Annual PM_10_ (µg/m^3^) 2018	Avg. Annual PM_2.5_ (µg/m^3^) 2018
Lancaster	144,426	(A6) Cable Street	~12,000	18/10/18	22	No data *
Birmingham	1,141,400	(A38) Bristol Road	~32,000	20/09/19	18	12
		Observation Site				
Mexico City	8,781,300	Constitución de la	~19,200	06/03/17	47	22
		República Avenue				

## Data Availability

All of the data is contained within the article and the Appendix A.

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
