# Peer review of "Oxidative Stress, Cytotoxic and Inflammatory Effects of Urban Ultrafine Road-Deposited Dust from the UK and Mexico in Human Epithelial Lung (Calu-3) Cells"

_antioxidants, 2022, doi:10.3390/antiox11091814_

Round 1

Reviewer 1 Report

Dear Editor,

After thoroughly reading the manuscript, I decided to reject it. Below are some comments on the manuscript that prevented me from accepting it even after potential corrections.

1.      The methods are described very carelessly. For example, ROS determination assay. While authors describe assay in main text in Materials and Methods additionally authors added paragraph on the subject in Supplementary methods that make difficult to understand.  In my opinion it would be much easier to follow Methodology part if methods would be combined in main text.

2.      References were divided in 2 parts. I have the same suggestion for this part of the manuscript. It would be much easier to follow manuscript having just one references list.

3.      Results.

a.      Figure3:  Authors show increase of proliferation or metabolic activity of cultures treated with UF-RDPs from Lancaster and suggest tumorigenic effect of Pb on this result. However they did not support this result with any additional data.  Moreover there are literature data showing proapoptotic effect of Pb on different cells.

b.      In Figure 4, the authors present the result of ROS levels in Calu-3 cells.  Again authors suggest that ROS levels induced by UF-RDPs of Lancaster were lower so they could stimulate proliferation. Additionally, authors show controls (TBHP treated cells) for sample from each city (labelled accordingly by different colours) separately. Does it mean that those controls were treated in different way?

c.      In figure 5 authors present cytokine levels in media of cells treated with UF-RDPs during 24h. I think it would be mor informative if authors would present cytokine levels giving real value of secreted IL6 and IL8 in pg/ml. Moreover when comparing % values of secreted IL6 and IL8 presented in this figure and supplementary figure S5 big discrepancies can be observed when comparing those 2 sets of results.  E.g.  increase of IL6 secretion (for samples spiked with UF-RDPs 300ug/ml) reach approx. 600% for particles from Lancaster, approx. 1800 for particles from Birmingham and approx. 1200 for particles from Mexico City in main text can not be compared to those in supplementary figure (even when viability of the cell is taken on account when calculating cytokine levels) Seems that completely different set of results are presented in those two figures.

d.      Finally, authors comment that level of LPS in UF-RDPs could not contribute to ROS generation in Calu-3 cells as they did not see any ROS production after Calu-3 exposure to high concentrations of LPS (10ug/ml) (line395-398). This is quite surprising as literature sources show ROS induction in cells treated with LPS even at concentration 1ug/ml.

Taking all these issues together I do not think that the manuscript has the quality expected for a Journal of high standards such as Antioxidants.

Reviewer 2 Report

·     I would suggest to the authors, in line 24, to briefly explain why worsening of the air is assumed.

·      In the introduction, it would be interesting to expand the bibliographical references, in particular, where the number of deaths due to human exposure to fine-grained airborne particles is indicated (mentioned by the authors in lines 34 and 35), if they have articles or reviews, in particular, more recent than 2015.

·       The sampling sites are well described, but the rationale for which they were chosen is unclear. It would be important, for the authors, to justify their choice appropriately.

·      The correct reference, made by the authors on the limits of the study relative to the difference in time, of the sampling phases, was very well done.
